# Deciphering the genetics and mechanisms of predisposition to multiple myeloma

Molly Went [1,25], Laura Duran-Lozano [2,3,25], Gisli H. Halldorsson [4], Andrea Gunnell [1], Nerea Ugidos-Damboriena [2,3], Philip Law [1], Ludvig Ekdahl [2,3], Amit Sud [1], Gudmar Thorleifsson [4], Malte Thodberg [2,3], Thorunn Olafsdottir [4], Antton Lamarca-Arrizabalaga [2,3], Caterina Cafaro [2,3], Abhishek Niroula [2,3], Ram Ajore [2,3], Aitzkoa Lopez de Lapuente Portilla [2,3], Zain Ali [2,3], Maroulio Pertesi [2,3], Hartmut Goldschmidt [5], Lilja Stefansdottir [4], Sigurdur Y. Kristinsson [6,7], Simon N. Stacey [4], Thorvardur J. Love [6,7], Saemundur Rognvaldsson [6,7], Roman Hajek [8], Pavel Vodicka [9], Ulrika Pettersson-Kymmer [10], Florentin Späth [11], Carolina Schinke [12], Frits Van Rhee [12], Patrick Sulem [4], Egil Ferkingstad [4], Grimur Hjorleifsson Eldjarn [4], Ulf-Henrik Mellqvist [13], Ingileif Jonsdottir [4], Gareth Morgan [14], Pieter Sonneveld [15], Anders Waage [16], Niels Weinhold [5,17], Hauke Thomsen [18], Asta Försti [17,19], Markus Hansson [2,20,21], Annette Juul-Vangsted [22], Unnur Thorsteinsdottir [4,7], Kari Hemminki [17,23], Martin Kaiser [1], Thorunn Rafnar [4], Kari Stefansson [4,7], Richard Houlston [1,26] ✉ & Björn Nilsson [2,3,24] ✉

Multiple myeloma (MM) is an incurable malignancy of plasma cells. Epidemiological studies indicate a substantial heritable component, but the underlying mechanisms remain unclear. Here, in a genome-wide association study totaling 10,906 cases and 366,221 controls, we identify 35 MM risk loci, 12 of which are novel. Through functional fine-mapping and Mendelian randomization, we uncover two causal mechanisms for inherited MM risk: longer telomeres; and elevated levels of B-cell maturation antigen (BCMA) and interleukin-5 receptor alpha (IL5RA) in plasma. The largest increase in BCMA and IL5RA levels is mediated by the risk variant rs34562254-A at *TNFRSF13B*. While individuals with loss-of-function variants in *TNFRSF13B* develop B-cell immunodeficiency, rs34562254-A exerts a gain-of-function effect, increasing MM risk through amplified B-cell responses. Our results represent an analysis of genetic MM predisposition, highlighting causal mechanisms contributing to MM development.

Multiple myeloma (MM) is one of the most common blood malignancies. It is defined by uncontrolled, clonal growth of plasma cells (Supplementary Fig. 1). Clinically, MM leads to bone marrow failure, bone lesions, and hypercalcemia and remains essentially incurable. It is preceded by monoclonal gammopathy of unknown significance (MGUS), a common condition (~3% of >50 year-olds) that progresses to MM at an annual rate of 1%.

First-degree relatives of MM and MGUS cases have a two- to four-fold higher risk for MM, as well as an increased risk for other B-cell malignancies and some solid tumors[1–4]. Genome-wide association

studies (GWAS) have identified DNA sequence variants at 25 loci influencing MM risk. However, much of the heritable risk remains unexplained[5–10], and the biological mechanisms involved are largely uncharacterized[11].

In the present study, we conducted a genome-wide association study totaling 10,906 MM cases and 366,221 controls. We identify 35 MM risk loci, 12 of which are novel. By integrating expression quantitative locus (eQTL), chromatin accessibility (ATAC-sequencing), and ultra-high-resolution chromatin configuration analysis (micro-C), we identify causal variants and high-confidence target genes. Using Mendelian Randomization analysis, we uncover two causal mechanisms for inherited MM risk: longer telomeres; and elevated levels of B-cell maturation antigen (BCMA) and interleukin-5 receptor alpha (IL5RA) in plasma. Moreover, we find that the largest increase in BCMA and IL5RA levels is mediated by the risk variant rs34562254-A at *TNFRSF13B* and that there is an antagonistic relationship between risk of B-cell immunodeficiency and risk of MM for this locus. Our results represent a comprehensive analysis of genetic MM predisposition, highlighting central biological mechanisms contributing to MM development.

## Results

### Genetic architecture of MM risk

To characterize the germline genetic architecture of MM, we performed a meta-analysis of ten GWAS datasets[5–10] totaling 10,906 cases

and 366,221 controls (Fig. 1a, Supplementary Data 1 and Supplementary Fig. 2). We identified 30 significant ($P_{meta} < 5 \times 10^{-8}$) and two suggestive associations ($P_{meta} < 5 \times 10^{-7}$), including nine novel significant associations (Supplementary Data 2). Approximate conditional analysis revealed three additional linkage disequilibrium (LD)-independent associations, yielding 12 novel associations (Fig. 1b and Table 1). We replicated all known associations except a previously reported borderline signal at 22q13.1/*TOM1* (rs138745; $P_{meta} = 0.001$)[7]. The two suggestive associations correspond to previously reported signals at 7q31.33/*POT1* and 6p22.3/*JARID2* ($P_{meta} = 7.1 \times 10^{-8}$ and $1.2 \times 10^{-7}$, respectively)[5,6]. MM plasma cell-specific transcriptome-wide association study (TWAS) and methylome-wide association study (MWAS) did not identify additional loci but provided support for 11 of the GWAS loci (Supplementary Data 3-4). Using linkage disequilibrium adjusted kinships (LDAK), we estimated the heritability ascribable to all common variation at 15.6% (± 4.7). Using LD score regression, we detected enrichment of risk variants in regions of accessible chromatin in plasma cells and B-cells (Fig. 1c), indicating that altered gene regulation in these cell types mediates MM risk. We also noted enrichment in activating histone marks of MM cell lines (Supplementary Fig. 3).

MM can be classified into hyperdiploid and non-hyperdiploid subtypes, the latter being primarily composed of cases with immunoglobulin heavy-chain (IGH) translocations, t(11;14), t(4;14) and t(14;16), which lead to over-expression of oncogenes, *CCND1*, *MMSET* and *MAF* respectively, through juxtaposition with the IGH locus.

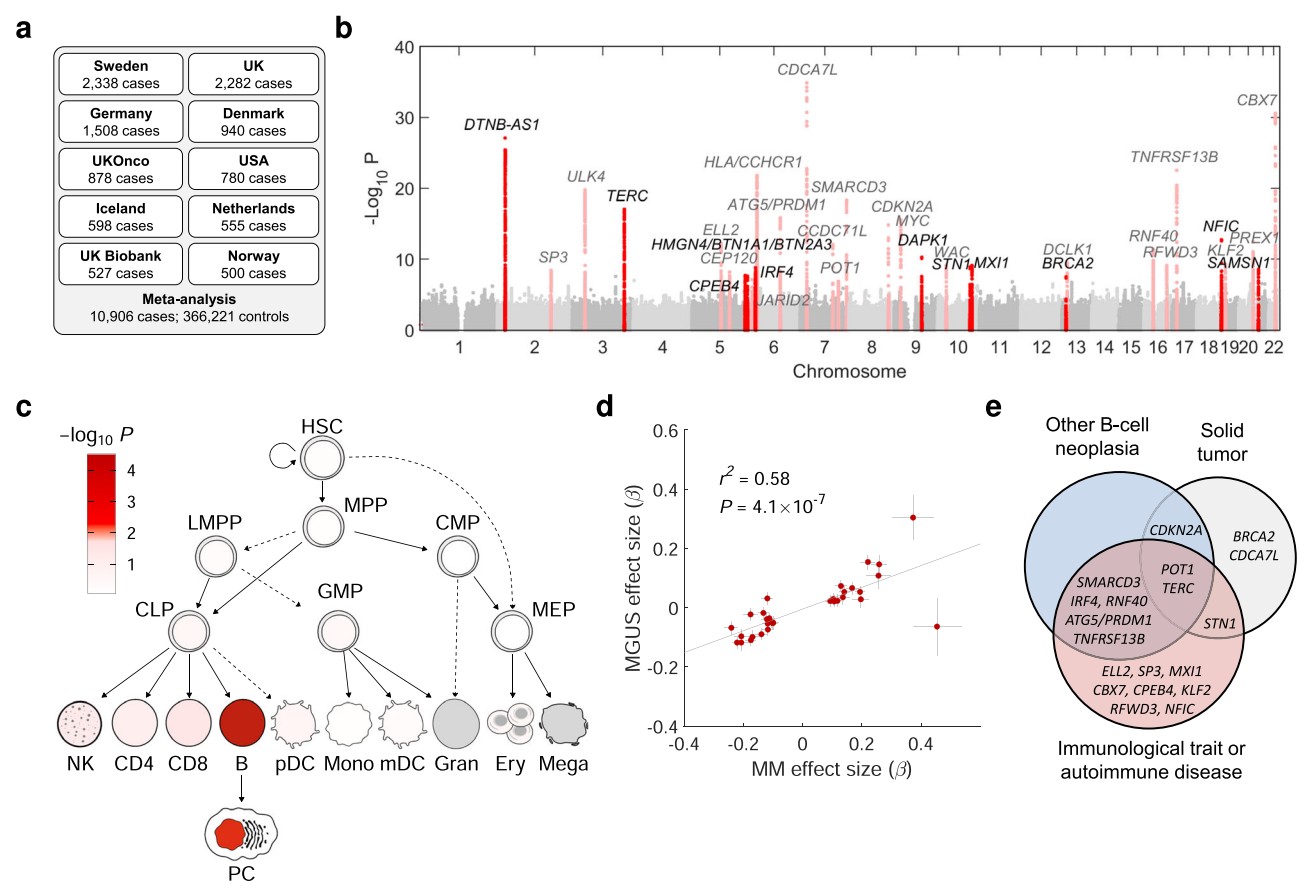

**Fig. 1 | Genetic architecture of MM risk. a** Study design. **b** Manhattan plot; *x*-axis indicates genomic position; *y*-axis −log₁₀ GWAS *P*-value. Dark red indicates loci where novel risk variants were found. **c** Enrichment of heritability in regions of accessible chromatin in hematopoietic cell types (red nuance indicates -log₁₀ LD-score regression *P*-value). **d** Correlation between MGUS and MM GWAS effect sizes (*β*) for the identified MM lead variants. *P*-value and *r²* statistics are for Pearson correlation. **e** Summary of additional pleiotropic associations (Supplementary

Data 7). Abbreviations: B-cells (B), CD4⁺ T-cells (CD4), CD8⁺ T-cells (CD8), common lymphoid progenitor (CLP), common myeloid progenitor (CMP), erythroid progenitor (Ery), granulocyte megakaryocyte progenitor (GMP), hematopoietic stem cells (HSC), lympho-myeloid primed progenitors (LMPP), monocyte (Mono), megakaryocyte-erythroid progenitor (MEP), multi-potent progenitors (MPP), myeloid dendritic cells (mDC), plasmacytoid dendritic cells (pDC), megakaryocyte (Mega), natural killer cells (NK), plasma cells (PC).

**Table 1 | Identified associations with MM risk**

| Cytoband | rsID | Chr | Position | RA/OA | RAF | OR | 95% CI | | | GWAS P | Phet | I² | Target genes |
|---|---|---|---|---|---|---|---|---|---|---|---|---|---|
| 7p15.3 | rs75341503 | 7 | 21936698 | A/C | 64.7 | 1.25 | 1.2 | - | 1.3 | 1.47E-35 | 0.94 | 0 | CDCA7L |
| 22q13.1 | rs5995688 | 22 | 39548027 | G/A | 43.8 | 1.21 | 1.2 | - | 1.3 | 2.71E-31 | 0.08 | 41 | CBX7 |
| 2p23.3 | rs7577599 | 2 | 25613146 | T/C | 76.4 | 1.27 | 1.2 | - | 1.3 | 8.00E-28 | 0.22 | 13 | DTNB-AS1 |
| **2p23.3\*** | **rs6546615** | **2** | **26148733** | **G/C** | **33.0** | **1.21** | **1.2** | **-** | **1.3** | **6.97E-22** | **0.09** | **40** | ***DTNB-AS1*** |
| 17p11.2 | rs34562254 | 17 | 16842991 | A/G | 10.6 | 1.30 | 1.2 | - | 1.4 | 2.82E-23 | 0.31 | 14 | TNFRSF13B |
| 6p21.33 | rs3132535 | 6 | 31116526 | A/G | 26.4 | 1.19 | 1.2 | - | 1.2 | 1.52E-22 | 0.14 | 33 | HLA, CCHCR1 |
| 3p22.1 | rs9856633 | 3 | 42013850 | A/G | 19.1 | 1.23 | 1.2 | - | 1.3 | 1.65E-20 | 0.44 | 0 | ULK4 |
| 7q36.1 | rs10233479 | 7 | 150933044 | T/C | 9.0 | 1.25 | 1.2 | - | 1.3 | 4.93E-19 | 0.15 | 33 | SMARCD3 |
| 3q26.2 | rs7621631 | 3 | 169512145 | C/A | 75.6 | 1.18 | 1.1 | - | 1.2 | 8.58E-18 | 0.43 | 1 | TERC |
| **3q26.2\*** | **rs77033531** | **3** | **169859690** | **G/C** | **98.0** | **1.63** | **1.4** | **-** | **1.9** | **1.72E-09** | **0.21** | **26** | ***TERC*** |
| 6q21 | rs9386514 | 6 | 106636902 | C/T | 19.2 | 1.18 | 1.1 | - | 1.2 | 1.51E-16 | 0.99 | 0 | ATG5, PRDM1 |
| 9p21.3 | rs3731222 | 9 | 21983914 | T/C | 85.2 | 1.23 | 1.2 | - | 1.3 | 2.84E-16 | 0.02 | 54 | CDKN2A |
| 8q24.21 | rs1948915 | 8 | 128222421 | C/T | 32.8 | 1.15 | 1.1 | - | 1.2 | 1.54E-15 | 0.32 | 13 | MYC |
| **19p13.3** | **rs11085015** | **19** | **3369572** | **T/G** | **17.3** | **1.19** | **1.1** | **-** | **1.3** | **1.73E-13** | **0.31** | **16** | ***NFIC*** |
| **19p13.3\*** | **rs8107139** | **19** | **3462045** | **C/T** | **39.0** | **1.13** | **1.1** | **-** | **1.2** | **5.11E-08** | **0.37** | **8** | ***NFIC*** |
| 20q13.13 | rs6090899 | 20 | 47358450 | G/A | 10.2 | 1.22 | 1.2 | - | 1.3 | 3.45E-13 | 0.26 | 20 | PREX1 |
| 5q15 | rs11744881 | 5 | 95240865 | A/T | 71.9 | 1.15 | 1.1 | - | 1.2 | 6.08E-13 | 0.37 | 8 | ELL2 |
| 7q22.3 | rs11762574 | 7 | 106293277 | A/G | 70.6 | 1.14 | 1.1 | - | 1.2 | 8.18E-13 | 0.89 | 0 | CCDC71L |
| 16p11.2 | rs8058928 | 16 | 30704312 | G/T | 28.6 | 1.14 | 1.1 | - | 1.2 | 3.82E-12 | 0.83 | 0 | RNF40 |
| **9q21.33** | **rs10746812** | **9** | **90099454** | **C/T** | **36.6** | **1.12** | **1.1** | **-** | **1.2** | **5.13E-11** | **0.97** | **0** | ***DAPK1*** |
| 13q13.3 | rs75712673 | 13 | 36766420 | G/T | 2.9 | 1.29 | 1.2 | - | 1.4 | 3.26E-10 | 0.55 | 0 | DCLK1 |
| 19p13.11 | rs4808046 | 19 | 16439390 | G/A | 23.0 | 1.13 | 1.1 | - | 1.2 | 4.62E-10 | 0.05 | 47 | KLF2 |
| 10p12.1 | rs2993984 | 10 | 28798656 | T/A | 73.7 | 1.12 | 1.1 | - | 1.2 | 7.32E-10 | 0.51 | 0 | WAC |
| **10q25.2** | **rs3737315** | **10** | **112035508** | **T/G** | **36.5** | **1.11** | **1.1** | **-** | **1.2** | **7.62E-10** | **0.25** | **21** | ***MXI1*** |
| 16q23.1 | rs8050262 | 16 | 74661159 | T/C | 59.3 | 1.11 | 1.1 | - | 1.2 | 7.83E-10 | 0.18 | 29 | RFWD3 |
| **10q24.33** | **rs11813268** | **10** | **105682296** | **T/C** | **15.5** | **1.15** | **1.1** | **-** | **1.2** | **1.30E-09** | **15.50** | **0** | ***STN1*** |
| **6p22.2** | **rs34565965** | **6** | **26350810** | **T/A** | **75.7** | **1.13** | **1.1** | **-** | **1.2** | **1.65E-09** | **0.17** | **30** | ***BTN1A1, BTN3A2, HMGN4*** |
| **21q11.2** | **rs2822736** | **21** | **15898681** | **C/G** | **38.5** | **1.11** | **1.1** | **-** | **1.2** | **2.79E-09** | **0.07** | **43** | ***SAMSN1*** |
| 2q31.1 | rs16862227 | 2 | 174832967 | G/T | 76.4 | 1.12 | 1.1 | - | 1.2 | 3.89E-09 | 0.22 | 24 | SP3 |
| 5q23.2 | rs2162826 | 5 | 122714477 | C/A | 21.9 | 1.12 | 1.1 | - | 1.2 | 6.58E-09 | 0.94 | 0 | CEP120 |
| **5q35.2** | **rs6864880** | **5** | **173298226** | **C/T** | **70.2** | **1.11** | **1.1** | **-** | **1.2** | **1.85E-08** | **0.91** | **0** | ***CPEB4*** |
| **6p25.3** | **rs1050976** | **6** | **408079** | **T/C** | **47.5** | **1.10** | **1.1** | **-** | **1.1** | **2.33E-08** | **0.49** | **0** | ***IRF4*** |
| **13q13.1** | **rs11571833** | **13** | **32972626** | **T/A** | **17.3** | **1.57** | **1.3** | **-** | **1.9** | **2.95E-08** | **0.31** | **0** | ***BRCA2*** |
| 6p22.3 | rs74875586 | 6 | 15216525 | A/G | 2.5 | 1.45 | 1.3 | - | 1.7 | 7.12E-08 | 0.12 | 36 | JARID2 |
| 7q31.33 | rs10954065 | 7 | 124672253 | C/A | 73.1 | 1.10 | 1.1 | - | 1.2 | 1.22E-07 | 0.81 | 0 | POT1 |

Novel loci in bold. Star (*) indicates conditional association. Abbreviations: *RA/OA* risk/other allele, *RAF* risk allele frequency, *OR* odds ratio, *95% CI* 95% confidence interval, *Phet* P-value for heterogenety; I2 heterogeneity, Q Cochran's Q.

Previous work has found relationships between the risk loci at 11q13.3/*CCND1* and 5q15/*ELL2* with t(11;14) and hyperdiploid MM, respectively[12,13]. For newly discovered loci, we found no evidence for additional subtype-specific associations (Supplementary Data 5).

To examine the genetic overlap with other diseases, we analyzed 6234 MGUS cases and 720,279 controls. We observed a strong, positive correlation with MGUS effect sizes for MM lead variants (Pearson $r^2 = 0.58$, $P = 4.2 \times 10^{-7}$; Fig. 1d and Supplementary Data 6), consistent with risk variants exerting their effects early in clonal evolution. Using the GWAS catalog[14], we also identified pleiotropy ($r^2 > 0.8$ between lead variants) with other B-cell neoplasias (8 signals), solid tumors (6 signals), autoimmune diseases (6 signals), and immunological traits (16 signals; Fig. 1e and Supplementary Data 7).

To assess the collective impact of all risk alleles, we calculated polygenic risk scores based on effect sizes and allele frequencies in our study population and the five super-populations in the 1000 Genomes compendium (European, American, African, East Asian, and South Asian). All identified risk variants were polymorphic in all super-populations, except the low-frequency variants at 6p22.3/*JARID2*, 7q36.1/*SMARCD3*, and 13q13.1/*BRCA2* which were not polymorphic in East Asians (Supplementary Data 8). Consistent with the higher incidence of MM among individuals of African or African-American ancestry, we observed the highest polygenic risk scores in the AMR super-populations (Supplementary Fig. 4; median 2.24 relative to our study population), due to a higher prevalence of several risk alleles (*e.g.*, 3p22.1/*ULK4*, 16p11.2/*RNF40*, 10q24.33/*STN1*, 19p13.3/*NFIC*, and 2p23.3/*DTNB-AS1*; Supplementary Data 8).

### Identification of target genes

To identify target genes, we considered genes overlapping a region defined by the variants in high LD ($r^2 > 0.8$) with the lead variant at each locus. Additionally, we considered genes with chromatin looping interactions with these regions, as determined by Micro-C analysis in MM cell lines. Among 371 genes in total, we prioritized target genes

based on (i) potentially pathogenic coding variants, (ii) variants in long non-coding RNAs, (iii) expression quantitative trait loci (eQTLs) in the B-cell lineage, and (iv) TWAS signals (Supplementary Data 3-9, 10). To identify putative causal variants underlying the effects on gene expression, we performed massively parallel reporter assays (MPRA) in three MM cell lines. We also incorporated published MPRA data[11], luciferase assays, and epigenomic annotations. Using conservative criteria (Online Methods), we identified 17 high-LD variants with transcriptional activity. Notably, 16 of these mapped to transcription start sites or enhancers (Table 2 and Supplementary Data 9).

In total, we identified 35 target genes (Fig. 2). Several of these were further supported by DepMap essentiality in MM or lymphoid cells (Supplementary Data 11), a Mendelian cancer predisposition syndrome (*BRCA2*, *CDKN2A*, *POT1*, and *RFWD3*)[15,16], a congenital B-cell immunodeficiency (*TNFRSF13B* and *WAC*)[17,18], or recurrent somatic genetic lesions in MM (*IRF4*, *MYC*, *PRDM1*, *JARID2*, *MXI1*, *TNFRSF13B*, and *POT1*)[19–23]. We also noted enrichment of target gene expression in the B-cell lineage (Supplementary Fig. 5). A more detailed description of all target genes is provided in Supplementary Notes.

## Biological pathways involved in MM predisposition
Pathway analysis showed that the set of target genes is enriched for genes involved in B-cell development, chromatin organization, and telomere maintenance (Supplementary Data 12). For example, *SAMSN1* encodes a regulator of B-cell activation, and *SAMSN1* deletions have been reported in MM-prone mice[24]. *TNFRSF13B* regulates B-cell homeostasis[25–31]. *ELL2* drives immunoglobulin (Ig) synthesis in plasma cells[32,33]. *PRDM1* and *ATG5* are essential for plasma cell survival[34]. Several other target genes interact with the MYC-IRF4 pathway, which plays a key role in B-cell and plasma cell development (Supplementary Fig. 6)[23,35]. These findings, and the enrichment of MM risk variants in accessible chromatin of plasma cells and B-cells (Fig. 1c), suggest that dysregulation of the germinal center and post-germinal center reaction is critical to MM predisposition.

Target genes involved in chromatin organization, cell cycle regulation, and DNA repair include *CDKN2A*, *RFWD3*[15], *NFIC*[36], *JARID2*[37], *SMARCD3*[11], *HMGN4*[38], and *CEP120*[39]. Notably, the 13q13.1 association represents a pathogenic truncating variant in *BRCA2* (Lys3326Ter)[40].

## Longer telomeres mediate genetic MM risk
Three target genes have well-known roles in telomere maintenance: *TERC* encodes the telomerase RNA component, *POT1* and *STN1* subunits of the shelterin complex. Given that leukocyte telomere length (LTL) is a marker for chromosomal instability[41–45], we assessed the pleiotropy between MM and LTL using data on 472,174 individuals from UK Biobank[46]. Using colocalization analysis, we found evidence of shared causal variants for increased MM risk and increased LTL at the *TERC*, *POT1*, and *STN1* loci (posterior probability, PP, of shared variant > 0.8; Fig. 3a and Supplementary Data 13). Additionally, using LDAK[47], we found a positive genetic correlation between MM and LTL ($R_g = 0.23$, $P = 1.87 \times 10^{-5}$).

To examine the causal effect of LTL on MM risk, we performed a two-sample Mendelian randomization analysis using four methods[48–50]. Increased LTL was consistently associated with increased MM risk (inverse variance weight random effects model *P*-value, $P_{IVW-RE} = 2.07 \times 10^{-4}$; Fig. 3b and Supplementary Data 14, 15), with the Steiger test confirming that this was the likely causal direction (Supplementary Data 16). These data support that a subset of risk variants increase MM risk by increasing telomere length, plausibly affecting replicative lifespan and/or chromosomal stability and thereby the risk of neoplastic transformation[51].

## Elevated plasma BCMA and IL5RA levels mediate genetic MM risk
To identify additional mechanisms underlying MM predisposition, we searched for shared effects of risk variants on B-cell and plasma cell development. These processes mainly take place in lymph glands and bone marrow. Since population-scale data is lacking for these tissues, we reasoned that shared mechanisms could be detectable indirectly through effects on circulating levels of proteins derived from these processes in peripheral blood.

Accordingly, we examined the effects of MM risk variants on the levels of 2931 plasma proteins using Olink data for 46,665 UK BioBank individuals. Across nine risk loci, we identified *trans*-protein quantitative trait loci (*trans*-pQTLs) for 21 proteins (Supplementary Data 17). Mendelian randomization analysis incorporating a Steiger test for directionality supported a causal relationship between increased levels of B-cell maturation antigen (BCMA; $P_{IVW} = 5.6 \times 10^{-6}$) and interleukin-5 receptor subunit alpha (IL5RA; $P_{IVW} = 9.0 \times 10^{-13}$) and increased MM risk (Fig. 3c-d). Both associations were replicated in SomaScan data for 36,177 Icelanders (Supplementary Data 16,18)[52]. Nine risk loci (*ATG5/PRDM1*, *CCHCR1*, *ELL2*, *MXI1*, *NFIC*, *RNF40*, *SMARCD3*, *TNFRSF13B* and *WAC*) showed significant association with BCMA and/or IL5RA. Colocalization analysis confirmed a shared variant with MM risk (PP > 0.8) at seven of these (*ELL2*, *MXI1*, *NFIC*, *RNF40*, *SMARCD3*, *TNFRSF13B* and *WAC*; Fig. 3a and Supplementary Data 19).

The BCMA receptor is expressed on plasma cells and mature B-cells. It binds B-cell activating factor (BAFF) and is a target for MM immunotherapy[53]. Its soluble form is produced by cleavage of the BCMA extracellular domain by γ-secretase[54]. Several studies have linked soluble BCMA levels to plasma cell pool size. For example, the plasma BCMA level decreases in MM patients after treatment, and patients with MGUS show lower levels than patients with fully developed MM[55–59]. IL5RA is also expressed in plasma cells and B-cells (Supplementary Fig. 7). IL5 stimulation promotes plasma cell differentiation and has been implicated in immunogenic MM cell death[60,61]. These results indicate that a second subset of MM risk variants exert their effects through increased BCMA and IL5RA levels, plausibly reflecting an expanded plasma cell and mature B-cell pool. These risk variants are distinct from those influencing telomere length (Fig. 3a).

## The *TNFRSF13B* risk variant predisposes for MM through a gain-of-function effect
To gain insight into the molecular basis of the elevated BCMA and IL5RA levels, we focused on the *TNFRSF13B* locus. The TNFRSF13B variant rs34562254-A is one of the most statistically significant MM risk variants. It is associated with the largest increase in BCMA and IL5RA levels ($P = 1.4 \times 10^{-97}$, $\beta = 0.23$ for BCMA; $P = 4.9 \times 10^{-63}$, $\beta = 0.19$ for IL5RA for rs34562254-A; Supplementary Data 17). In addition, we and others have demonstrated an association between rs34562254-A and higher IgG levels[62–64].

*TNFRSF13B* encodes the TACI receptor, a central regulator of B-cell responses and Ig class-switching. Individuals who carry rare loss-of-function variants in *TNFRSF13B* are predisposed to common variable immunodeficiency (CVID), a condition defined by low IgG and IgA levels due to stalled development of mature B-cells and plasma cells[17]. The most common CVID variants in *TNFRSF13B* are the Cys104Arg and Ala181Glu missense variants, which abolish TACI signalling[65]. We found associations between Cys104Arg and Ala181Glu and lower BCMA and IL5RA levels in the UK Biobank Olink data (Fig. 4a and Supplementary Data 22). The opposite effects on BCMA, IL5RA, and IgG levels shown by rs34562254-A indicate that this MM risk variant has a gain-of-function effect.

Searching for putative causal variants, we noted that rs34562254 is a missense variant (Pro251Leu) that is predicted to be benign[66,67]. However, we noted an association between rs34562254-A and increased *TNFRSF13B* expression in B-cells (Supplementary Data 10) and, congruent with this, two variants in high LD (rs4273077 and rs4792800; $r^2 = 0.90$ and 0.92 with rs34562254) showed transcriptional effects in both MPRA datasets (Fig. 4b-c, Table 2 and Supplementary Data 9). Further, both rs4273077 and rs4792800 map to

## Table 2 | Putative causal variants in high LD with MM lead variants

| MM lead variant | $r^2$ | Variant | Target gene | RA/OA | Genetic effect | Effect in reporter assay | Regulatory elements |
|---|---|---|---|---|---|---|---|
| rs10233479 | 0.91 | rs73169649 | SMARCD3 | C/T | Increasing expression | Increasing activity[c] | - |
| rs10233479 | 1.00 | rs78740585 | SMARCD3 | A/G | Increasing expression | Increasing activity[c,d] | Enhancer in SMARCD3; Looping to SMARCD3 TSS |
| rs10746812 | 0.77 | rs1329600 | DAPK1 | G/A | Increasing expression | Increasing activity[e] | DAPK1 TSS |
| rs11571833 | 1.00 | rs11571833 | BRCA2 | T/A | Lys3326Ter[a] | - | - |
| rs11744881 | 0.96 | rs1458018 | ELL2 | G/T | Decreasing expression | Decreasing activity[c] | Enhancer in ELL2 |
| rs11744881 | 0.96 | rs17085266 | ELL2 | A/C | Decreasing expression | Decreasing activity[c] | Enhancer in ELL2 |
| rs11744881 | 0.91 | rs3777182 | ELL2 | T/A | Decreasing expression | Decreasing activity[c,d] | Enhancer in ELL2 |
| rs11744881 | 0.90 | rs3777183 | ELL2 | G/A | Decreasing expression | Decreasing activity[d] | Enhancer in ELL2 |
| rs11744881 | 0.96 | rs3777189 | ELL2 | C/G | Decreasing expression | Decreasing activity[d] | Enhancer in ELL2 |
| rs11744881 | 0.91 | rs889302 | ELL2 | A/C | Decreasing expression | Decreasing activity[c] | Enhancer in ELL2 |
| rs2822736 | 1.00 | rs2822736 | SAMSN1 | G/A | Increasing expression | Increasing activity[e] | Enhancer in SAMSN1 |
| rs2993984 | 0.88 | rs2790444 | WAC | C/T | Decreasing expression | Decreasing activity[d] | Enhancer in WAC |
| rs34562254 | 1.00 | rs34562254 | TNFRSF13B | A/G | Pro251Leu[b] | - | - |
| rs34562254 | 0.96 | rs4273077 | TNFRSF13B | G/A | Increasing expression | Increasing activity[c,d] | Enhancer in TNFRSF13B; Looping to TNFRSF13B |
| rs34562254 | 0.92 | rs4792800 | TNFRSF13B | G/A | Increasing expression | Increasing activity[c,d] | Enhancer in TNFRSF13B; Looping to TNFRSF13B TSS |
| rs34565965 | 1.00 | rs34565965 | BTN1A1, BTN3A2, HMGN4 | T/A | Increasing BTN3A2, HMGN4; Decreasing BTN1A1 | Increasing activity[e] | Enhancer 14.7 kb upstream of BTN1A1; Looping to BTN1A1 and HMGN4 |
| rs6864880 | 0.87 | rs144869372 | CPEB4 | CCCTTCG/C | Decreasing expression | Decreasing activity[e] | Enhancer 4.9 kb upstream of CPEB4; Looping to CPEB4 TSS |
| rs6864880 | 0.82 | rs72810983 | CPEB4 | A/G | Decreasing expression | Decreasing activity[e] | CPEB4 TSS |
| rs75341503 | 0.94 | rs4487645 | CDCA7L | C/A | Increasing expression | Increasing activity[d] | Enhancer 3.6 kb downstream CDCA7L |
| rs7621631 | 0.81 | rs2293607 | TERC | T/C | Variant in long non-coding RNA | - | - |

[a]Truncating variant.
[b]Missense variant in signaling domain. Effect directions with respect to MM risk allele.
[c]New MPRA.
[d]Published MPRA.
[e]Luciferase.

# a

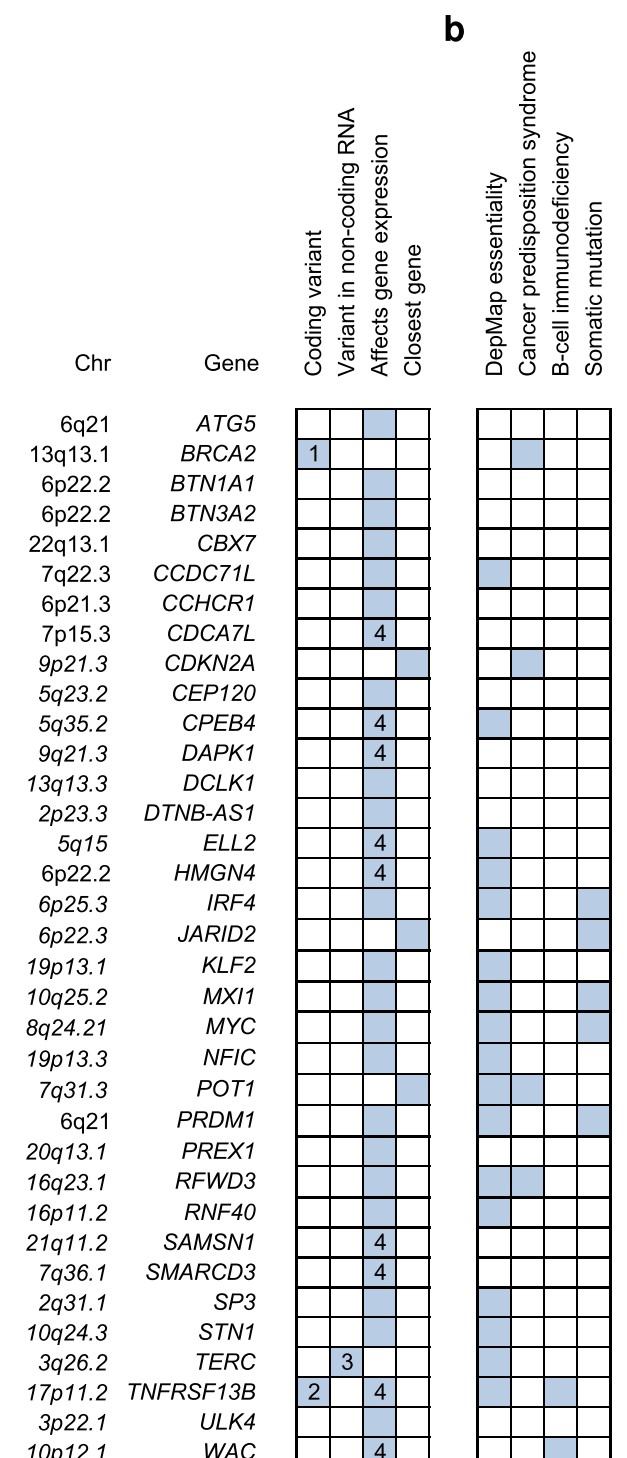

# b

**Fig. 2 | Overview of target genes. a** Among genes located in associated regions, or with chromatin looping contact with these regions, we prioritized target genes based on highly correlated ($r^2 > 0.8$) coding variants, variants in long non-coding RNAs, eQTLs, and TWAS signals (Supplementary Data 3,10). Footnotes: 1: Truncating variant (Lys3326Ter). 2: Missense variant (Pro251-Leu) in intracellular signalling domain. 3: Variant in expressed sequence of *TERC* (rs2293607; $r^2 = 0.81$ with rs7621631). 4: eQTL supported by a transcriptionally active variant (Table 2 and Supplementary Data 9). **b** Additional support for target genes, including DepMap essentiality (Supplementary Data 11), cancer predisposition syndromes, B-cell immunodeficiencies, or somatic mutations in MM.

enhancers in *TNFRSF13B* intron 3, and rs4792800 displays a chromatin looping interaction with the transcription start site (Fig. 4b and Supplementary Data 9). Finally, dual-sgRNA CRISPR/Cas9 deletion of the rs4273077- and rs4792800-harboring regions in Raji cells led to the downregulation of both of the two main *TNFRSF13B* transcript isoforms (Fig. 4d), further supporting a regulatory role of these regions. These data indicate that the *TNFRSF13B* MM risk allele exerts a gain-of-function effect leading to increased MM risk.

## Discussion

We report a comprehensive analysis of the germline genetic architecture of MM. By bringing together all major GWASs to date, we identify 12 new risk loci. Through functional fine-mapping, we identify high-confidence target genes and central biological pathways (Fig. 5 and Supplementary Notes). Our data support that MM risk variants act early in clonal evolution by predisposing for MGUS rather than for progression from MGUS to MM.

Furthermore, we identify two central mechanisms mediating inherited MM risk: increased LTL and increased BCMA and IL5RA levels. These findings are consistent with a model where dysregulation of telomere maintenance and B-cell and plasma cell development constitute central mechanisms in MM predisposition, each influenced by a distinct subset of risk loci (Fig. 3a). Our initial analysis of the *TNFRSF13B* risk locus suggests that the increase in BCMA and IL5RA levels reflects a gain-of-function effect leading to increased MM risk through amplified B-cell responses (Fig. 4a).

In conclusion, our study provides insights into genetic MM predisposition, highlighting central biological mechanisms that lead to MM.

## Methods

### Ethics

Collection of patient samples and clinico-pathological information was undertaken with informed consent and ethical approvals in accordance with the Declaration of Helsinki: for the Myeloma-IX[68,69] trial by the Medical Research Council Leukaemia Data Monitoring and Ethics committee (MREC 02/8/95, ISRCTN68454111), the Myeloma-XI[70] trial by the Oxfordshire Research Ethics Committee (MREC 17/09/09, ISRCTN49407852), HOVON65/GMMG-HD4 (ISRCTN 644552890; METC 13/01/2015), HOVON87/NMSG18 (EudraCTnr 2007-004007-34, METC 20/11/2008), HOVON95/EMN02 (EudraCTnr 2009-017903-28, METC 04/11/10), University of Heidelberg Ethical Commission (229/2003, S-337/2009, AFmu-119/2010), University of Arkansas for Medical Sciences Institutional Review Board (IRB 202077), Lund University Ethical Review Board (2022-01414-02), the Norwegian REK 2014/97, the Danish Ethical Review Board (no. H-16032570), and the National Bioethics Committee of Iceland (VSN 17-143).

### Data reporting

No statistical methods were used to predetermine sample sizes. Experiments were not randomized, and the investigators were not blinded.

### Genome-wide association study

We performed a meta-analysis of ten GWAS data sets from nine previously published studies (German, US, UKOnco, UK, Netherlands, Sweden, Norway, Denmark, and Iceland)[5–10] and the UK Biobank (UKBB), totalling 10,906 cases and 366,221 controls, all population–based cohorts with European Ancestry. Published studies: The nine GWAS comprised Swedish (2338 cases, 11,971 controls), UK (2282 cases, 5197 controls), German (1508 cases, 2107 controls), Danish (940 cases, 91,744 controls), UKOnco (878 cases, 7083 controls), US (780 cases, 1857 controls), Netherlands (555 cases, 2669 controls), Icelandic (598 cases, 313,882 controls), and Norwegian (500 cases, 4696 controls) and. UK Biobank study[71]: 527 cases of MM and 1417 age

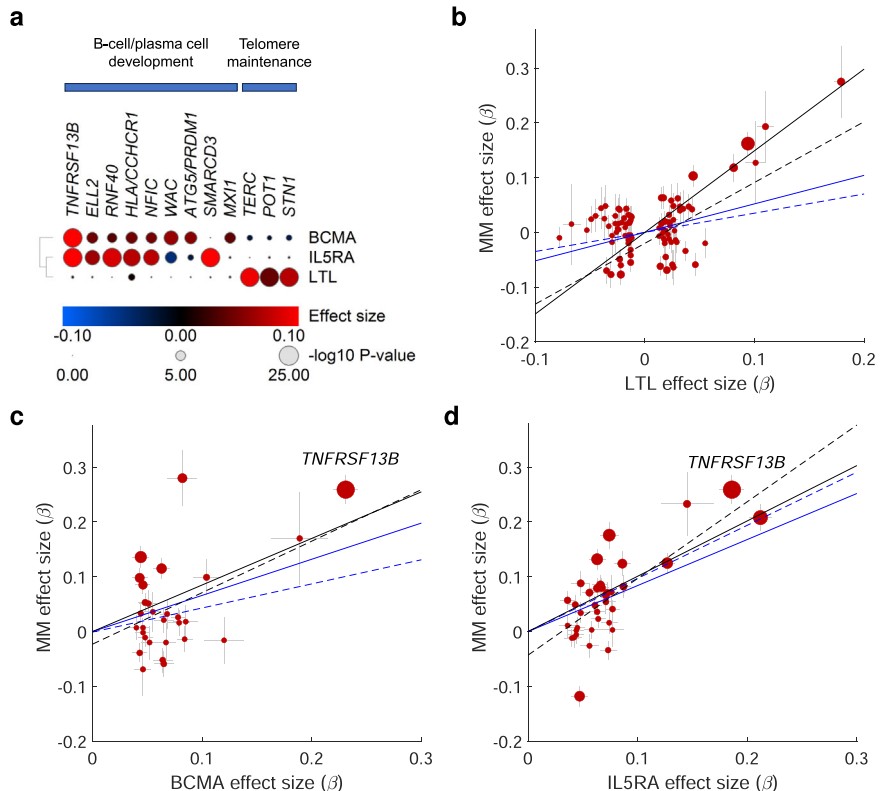

**Fig. 3 | Pleiotropy with LTL and plasma BCMA/IL5RA levels. a** MM risk variants showing colocalized associated with LTL, BCMA levels, or IL5RA levels in UK Biobank (Supplementary Data 13,19). Color indicates effect size ($\beta$) with respect to the MM risk variant. Marker size indicates -log$_{10}$ GWAS *P*-value for association with the respective traits. **b**–**d** Mendelian randomization (MR) plots showing effect sizes ($\beta$) of LTL, BCMA, and IL5RA GWAS variants in the UK Biobank (exposures) and their effect sizes ($\beta$) on MM risk (outcome). Lines represent slopes of four tests: inverse-variance weighted (blue solid), weighted median (blue dashed), weighted mode (black solid), and MR-Egger (black dashed). Data represent effect size ($\beta$) ± s.e.m; circle area -log$_{10}$ GWAS *P*-value.

and sex-matched controls were identified, and genotypes downloaded. The association between variant genotype and MM was performed under an additive model in SNPTESTv2.5. The diagnosis of MM (ICD-10 C90.0) was established in accordance with World Health Organization guidelines. All samples from patients for genotyping were obtained before treatment or at presentation.

We examined the relationship between genotype and MM in each GWAS, assuming a log-additive model[72]. Meta-analysis under a fixed-effects inverse-variance weighted model was performed using META v1.7[73]. Variants in the meta-analysis only included those with an imputation quality score (info) > 0.8 and MAF > 0.005 (8.1 million variants after filtering). The $I^2$ statistic was calculated to quantify between-study heterogeneity, and variants with an $I^2 > 75\%$ were excluded. There was no evidence of genomic inflation ($\lambda = 1$, Supplementary Fig. 1). To define known risk loci, we compiled a list of previously reported genome-wide significant association signals for MM (*i.e.*, $P < 5 \times 10^{-8}$). Genome-wide complex trait analysis was used to perform approximate conditional and joint association analysis (COJO)[74] to identify independent risk loci. To estimate LD, we used a reference sample of unrelated individuals from a combined dataset of UK10K[75] and European individuals from the 1000Genomes Project[76], excluding variants with low imputation quality (INFO < 0.8) and deviation from HWE ($P < 1 \times 10^{-6}$). Associations at $P_{conditional} < 5 \times 10^{-8}$ within a 1 Mb region of primary associations were considered novel secondary associations.

**Transcriptome-wide association study**
We retrieved previously published expression data generated on CD138-purified plasma cells from 183 UK (MRC Myeloma IX trial, GSE21349), 658 German (E-MTAB-2299), and 608 US cohorts (GSE2658, GSE31161)[77]. RNA was profiled using Affymetrix Human Genome U133 2.0 Plus Arrays. Gene expression models were generated using the PredictDB pipeline[78] for a total of 1449 participants. Elastic net model building was done independently for each dataset. Models were computed using genotype and expression data, and covariate factors were estimated using PEER[79]. For the UK dataset, 30 PEER factors were used; for the US and German data sets, 60 PEER factors were used, as recommended by the GTEx protocol. Transcriptome-wide association tests were performed for each dataset individually using S-PrediXcan[80] with summary statistics from the GWAS meta-analysis. To combine S-PrediXcan results from the different data sets, we used S-MultiXcan[81].

**Methylome-wide association studies**
Illumina 450 K methylation array data was obtained from 379 of the UK cohort (MRC Myeloma XI trial). The EZ DNA Methylation kit (Zymo Research) was briefly used for bisulfite conversion of genomic DNA. Tumour DNA methylation was profiled using Illumina Infinium HumanMethylation450 arrays. Raw data were exported from Genome Studio (Illumina). Quality checking and normalization of raw methylation data on 378 cases was performed using the ChIP Analysis Methylation Pipeline (ChAMP). The BMIQ method was used to perform normalization. Elastic net model building was performed using genotype and expression data and covariate factors estimated using PEER, where 60 PEER factors were according to the GTEx protocol. Methylome-wide association tests were then performed for the dataset using S-PrediXcan with summary statistics from the GWAS meta-analysis. We annotated CG islands with the nearest gene and considered a Bonferroni-corrected *P*-value of $2 \times 10^{-6}$ (*i.e.*, 0.05/25,000 genes) as significant.

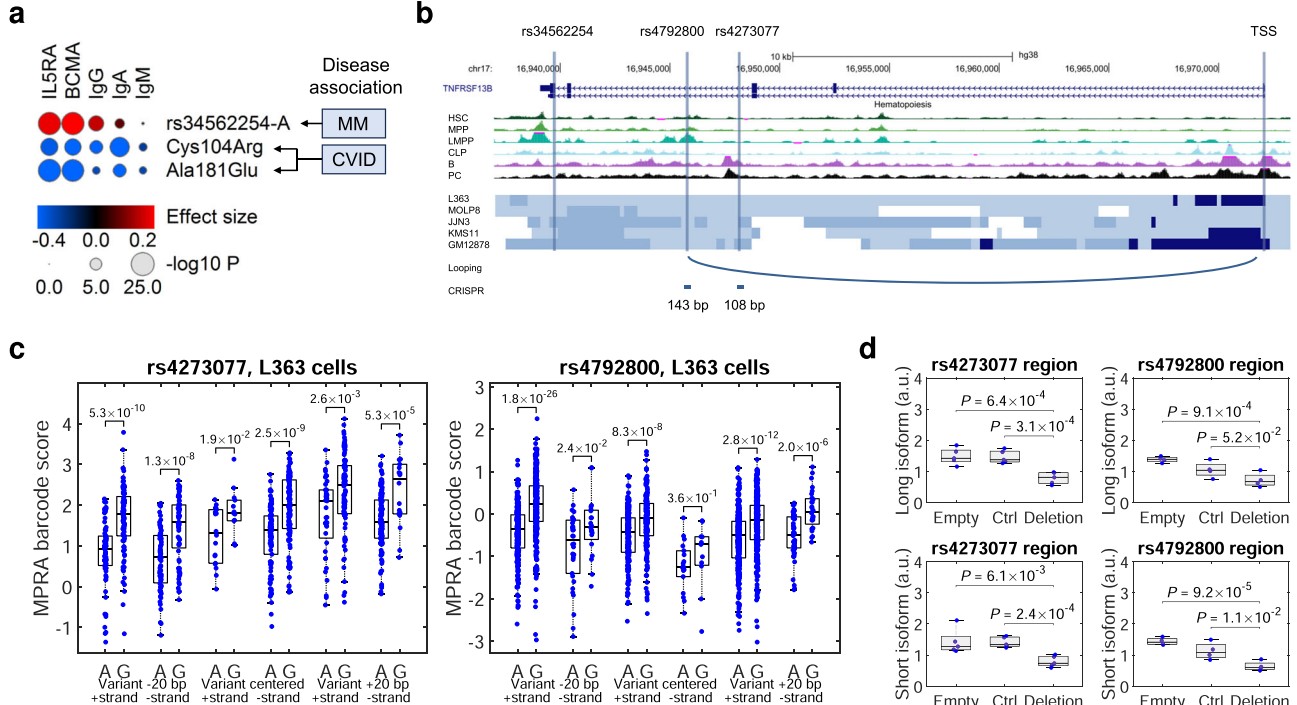

**Fig. 4 | Functional fine-mapping of the *TNFRSF13B* locus. a** Effects of the MM lead variant rs34562254-A on BCMA, IL5RA, IgG, IgA, and IgM levels in the UK Biobank. Also shown are the Cys104Arg and Ala181Glu loss-of-function variants associated with CVID (Supplementary Data 22). **b** Genomic context of rs34562254, rs4273077, and rs4792800, showing chromatin accessibility (ATAC-seq intensity) in the B-cell lineage, looping interactions, and regions targeted by CRISPR/Cas9. Also shown are the chromatin states identified through ChromHMM analysis of histone mark ChIP-seq data for in four plasma cell line (L363, MOLP8, JJN3, and KMS11) and one B-cell line (GM12878). Light blue indicates enhancer activity; medium blue transcriptionally active chromatin; dark blue transcription start site; and white indicates transcriptionally inactive/repressed chromatin. **c** MPRA data for rs4273077 and rs4792800 in L363 cells. Dots represent effect estimates for individual MPRA barcodes (MPRAscore $\beta_i$ values reflecting the representation of a barcode at the RNA level normalized to its representation in the MPRA plasmid library), grouped by allele (reference allele to the left; alternative to the right), DNA strand (+ or -) and sliding window (variant at −20, 0 or +20 bp from the center of 120 bp oligonucleotides representing the genomic context). Overall MPRA signals for each cell line in Supplementary Data 9. **d** Expression of long and short *TNFRSF13B* isoforms in Raji cells subjected to dual-sgRNA CRISPR/Cas9 deletion of the rs4273077- and rs4293800-harboring regions ("CRISPR"), non-targeting control ("Ctrl"), or empty vector ("Empty"). *P*-values are for Student's *t*-test. The bottom, middle and top of each box plot represent the 25:th, 50:th, and 75:th percentiles. The whiskers represent the non-outlier minimum and maximum values, located at 1.5 times the interquartile range from the bottom and top of the box, respectively. The numbers by the brackets are *P*-values for two-sided Student's *t*-test.

## Estimation of heritability and partitioned heritability

To estimate the narrow-sense heritability of MM risk, we used LDAK v5.2, applying BLD-LDAK and LDAK-Thin models[47]. Variants were harmonized to HapMap3 with 1000 Genomes EUR, MAF > 0.01. Transformation of observed scale heritability estimates of MM to the liability scale was carried out, assuming a lifetime risk of 1% for MM. To estimate cell type-specific partitioned heritability based on chromatin accessibility, we used LD-scores based on ATAC-seq data for sorted blood cells available for LDSC[12], extended with LD-scores for myeloid, plasmacytoid dendritic cells, and plasma cells computed from published ATAC-seq data[8] (NCBI Gene Expression Omnibus accession no. GSE119453; European Genome-phenome Archive accession no. EGAS00001005394 and EGAD00001007814).

## ChIP-mentation and ATAC-seq data

We carried out ChIPmentation and ATAC-seq to annotate regulatory elements in KMS11 cells[82–84]. ChIPmentation was carried out for histone marks H3K27Ac, H3K27me3, H3K4me1, H3K4me3, H3K36me3 and H3K9me3 in KMS11, L363, JJN3 and MOLP-8 cell lines. ChIPmentation reads were trimmed and aligned to hg19/GRCh37 using Bowtie2. Duplicate reads were marked and removed using Picard. ChromHMM was used to infer chromatin states, training the model on four cell lines. Genome-wide signal tracks were binarized, including input controls. A 12-state model was assigned to the states[85–87]. We also annotated variants using GM12878 and Bone

Marrow mesenchymal stem cell ChromHMM tracks, using Roadmap Epigenomics data[88,89].

## Cell culture

KMS11, KMS12-BM, L363, MOLP8, MM.1 S, U266B1, and Raji cells were obtained from ATCC, cultured under recommended conditions, and tested for mycoplasma.

## Variant set enrichment analysis in ChIP-seq data

To examine enrichment in binding across risk loci, we adapted the method of Cowper-Sal lari et al.[90]. Briefly, for each risk locus, a region of strong LD (defined as $r^2 > 0.8$ and D′ > 0.8) was determined, and these variants were considered the associated variant set (AVS). ChIP-seq peak data for six histone marks from KMS11, L363, MOLP8, and JJN3 cell lines were generated in-house. For each mark, the overlap of the variants in the AVS and the binding sites was assessed to generate a mapping tally. A null distribution was produced by randomly selecting variants with the same characteristics as the risk-associated variants, and the null mapping tally was calculated. This process was repeated 10,000 times, and *P*-values were calculated as the proportion of permutations where the null mapping tally was greater or equal to the AVS mapping tally. An enrichment score was calculated by normalizing the tallies to the median of the null distribution. Thus, the enrichment score is the number of standard deviations of the AVS mapping tally from the median of the null distribution tallies.

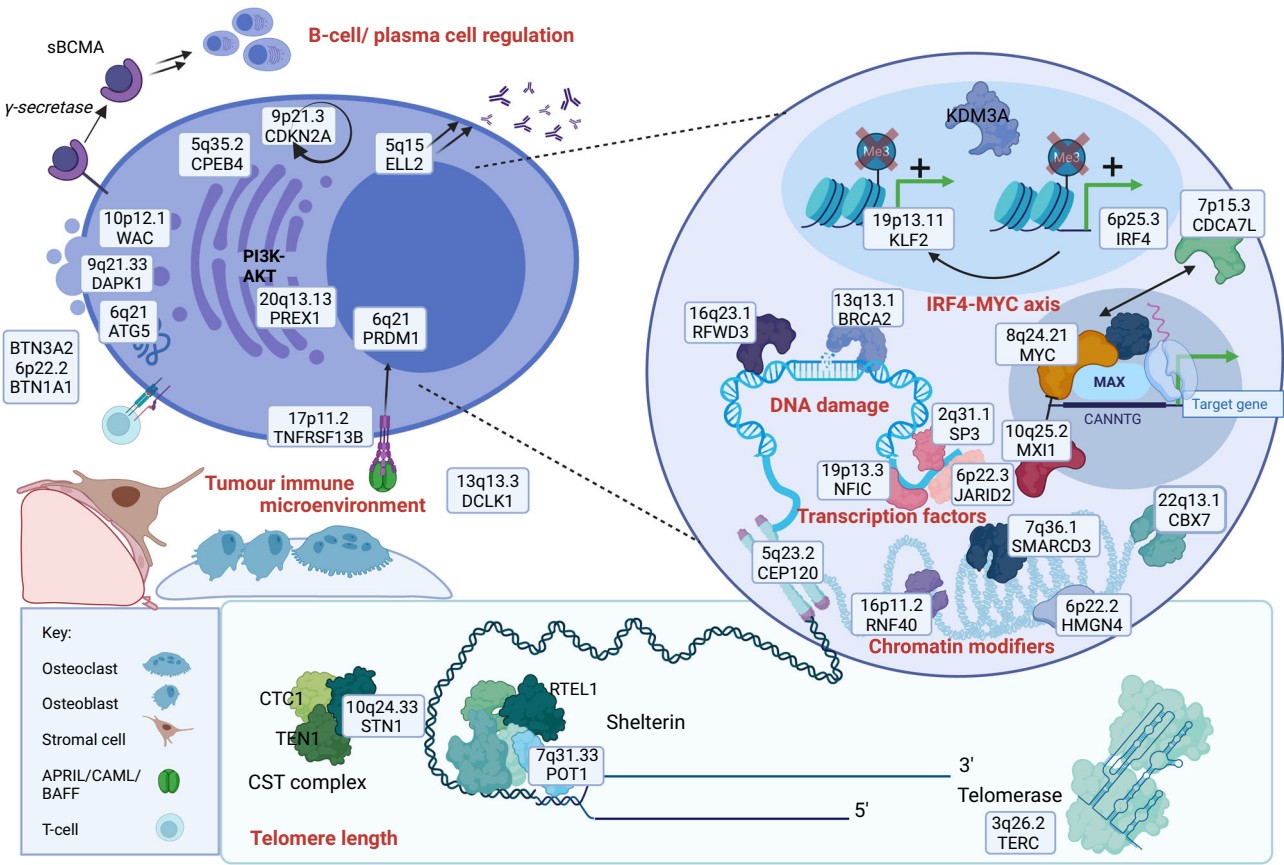

**Fig. 5 | Genes and pathways involved in MM predisposition.** The boxes indicate target genes and the locus at which they were identified. A description of each locus and target gene is provided in Supplementary Notes.

## Association data for MGUS

Summary statistics for the MM variants were obtained from a meta-analysis of 6234 MGUS cases and 720,297 controls from Iceland (4092 cases and 298,673 controls), UK Biobank (1150 cases; 427,714 controls) and the German Cancer Research Center (992 cases; 2910 controls). Cases were defined based on ICD-10 code D47.2. The Icelandic samples were genotyped using Illumina microarrays UK Biobank samples using Affymetrix UK BiLEVE / UK Biobank Axiom chips. Genotypes were long-range phased and imputed using population-specific reference sets[91] (whole-genome sequencing data for 63,118 Icelandic and 150,119 UK Biobank individuals). The association between variants and MGUS was tested using logistic regression assuming an additive model[92]. Association results for individual cohorts were meta-analyzed using inverse variance weighted meta-analysis.

## Pleiotropy analysis

We used the GWAS catalog (accessed November 2023) to identify pleiotropy with other diseases, as well as with hematological and immunological traits. We considered highly correlated associations ($r^2 > 0.8$ between the GWAS catalog and MM lead variants).

## Micro-C analysis

To detect chromatin looping interactions between risk variants and target genes, we carried out Micro-C analysis as per refs. 93,94 with the following modifications: Cells were fixed at a density of 106 cells per ml in 3 mM disuccinimydyl glutarate (DSG) for 20 mins at room temperature (RT). After adding formaldehyde to a final concentration of 1%, cells were further incubated at room temperature for 10 mins. Reactions were quenched by adding glycine to a final concentration of 660 mM with incubation for 5 mins at RT. Fixed cells were digested

with MNase (Worthington) optimized for each cell line and batch, with incubation for 10 mins at 37 °C, 1000 rpm. The reaction was quenched by EGTA at a final concentration of 12.5 mM, with incubation for 10 mins at 65 °C, 1000 rpm. End repair and biotin labeling was performed by incubating 106 MNase-digested cells using 30U of T4 PNK (NEB) at 37 °C for 15 mins, 1000 rpm. 35 U Large Klenow Fragment (NEB) was added and incubated at 37 °C for 15 mins. After biotin 14-dATP (Jena NU-809-BIOX), biotin 11-dCTP (Jena NU-835-BIO14), dTTP and dGTP were added (final concentration 66uM each), samples were incubated at 25 °C for 45 mins, 1000 rpm. Reactions were stopped using 40 mM EDTA and heating to 65 °C for 20 min. Ligation was performed with 10,000 U T4 DNA ligase, 23 °C for 3 hours, 1000 rpm. Biotin ends were excised using 200U Exonuclease III at 37 °C for 10 mins, 1000 rpm. Sequencing was conducted using a NovaSeq (Illumina). The Juicer (Aidenlab) pipeline was used to generate Hi-C maps from raw fastq files, and Mustache and FitHiC2 (Ay-lab) were used to call significant interactions.

## Expression quantitative locus (eQTL) data sets

To identify cis-eQTLs in plasma cells, we analyzed gene expression profiles of CD138+ cells isolated from bone marrow aspirates from MM patients harvested using immunomagnetic beads. First, we used Affymetrix microarray data for 1445 subjects, including 183 UK Myeloma IX trial patients (a study aimed at comparing two bisphosphonates in the treatment of MM; Medical Research Council Leukaemia Data Monitoring and Ethics committee, no. MREC 02/8/95, ISRCTN68454111)[95], 658 German GMMG patients, and 604 patients treated at the University of Arkansas for Medical Sciences Myeloma Center, USA[6]. Second, we used 185 RNA-seq samples from Lund University (Lund, Sweden)[96]. Third, we used 758 RNA-seq samples with

DNA copy-number covariates from the CoMMpass study[97]. Fourth, we used 309 RNA-seq samples from the Dana Farber Cancer Institute (Boston, USA)[98]. For the first two data sets, paired SNP microarray genotypes were available. For the third and fourth data sets, only RNA-seq data were available, limiting eQTL analysis to risk alleles with these coding proxies. Additionally, we used mRNA-sequencing data for 28 sorted immune cell populations from 416 individuals from the ImmunExUT compendium[99]. For B cells, we used eQTL data for 758 Icelanders generated by isolating B-cells from peripheral blood through negative selection using magnetic beads (StemCell Technologies 19674). We used eQTL data from deCODE Genetics (RNA-seq for 17,848 Icelanders) for whole blood.

### Selection of target genes and putative causal variants

To identify target genes underlying the MM risk associations, we considered genes overlapping a region defined by the variants in high LD ($r^2 > 0.8$) with each lead variant. Additionally, we considered genes displaying chromatin looping interactions with these regions, as determined by the Micro-C data. Among 371 admissible genes in total, we prioritized genes as probable target genes if they: (i) contained a potentially pathogenic coding variants correlated ($r^2 > 0.8$) with the MM lead variant; (ii) contained a variant in the expressed sequence of a long non-coding RNAs correlated ($r^2 > 0.8$) with the MM lead variant; (iii) had expression quantitative trait loci (eQTLs) in plasma cells or another B-cell population ($r^2 > 0.8$ between the eQTL and MM lead variant); or (iv) a Bonferroni-significant TWAS signal within 1 Mb of the MM lead variant. We considered potentially pathogenic variants as frameshift, stop-gain, stop-loss, and splice variants; computationally predicted-pathogenic missense variants, and missense variants in functionally well-characterized protein domains. At loci where no effect on expression could be identified in the B-cell lineage, we accepted eQTLs in other hematologic cell populations. At loci, where no gene fulfilled any of our criteria, we prioritized the closest gene.

We searched for putative causal gene-regulatory variants to obtain mechanistic support for the identified effects on gene expression. In the newly generated MPRA data (KMS11, RPMI-8226, and L363 cells), we nominated variants with false discovery rate (FDR) < 5% in at least two cell lines and absolute $\log_2$ fold-change >0.2 in at least one of these. In the published MPRA data (L363 and MOLP8 cells), we nominated variants with FDR < 5% in both cell lines and absolute $\log_2$ fold-change >0.2 in at least one of them. In addition, we nominated variants with significant effects in luciferase assays. We only considered effects in the same direction as the corresponding eQTL/TWAS signal.

To obtain further support a functional impact of the gene itself, we examined effects of CRISPR/Cas9 and shRNA knockdown on MM and lymphoid cell line growth using data from the Dependency Map (DepMap; https://www.depmap.org; version 23Q2), associations with human Mendelian cancer predisposition syndrome, congenital B-cell immunodeficiencies, and occurrence of recurrent somatic mutations in MM.

### Massive parallel reporter assays

In addition to making use of our published MPRA data for the L363 and MOLP8 MM cell lines for 21 risk loci[11], we generated an expanded MPRA dataset for 23 of the risk loci using the KMS11, RPMI8226, and L363 MM cell lines[100]. Single-base pair variants in LD ($r^2 \geq 0.4$) of the lead variant were included at each locus. Candidate regulatory sequences (CRS) were designed in the forward and reverse direction for reference (ref) and alternate (alt) alleles. Variants were centered in a 200-bp region. 230-bp oligos were synthesised (Agilent) with the CRS between 15 bp adapters- ACTGGCCGCTTGACG**CRS**CACTGCGGCTCCTGC. Two rounds of PCR were used to add a minimal promoter (primers 5BC-AG-f01v2 and 5BC-AG-r01v2; Supplementary Data 23) and a 15 bp random barcode. Amplified fragments were cloned by Gibson assembly into the SbfI/AgeI site of the pLS-SceI vector (Addgene no. 137725) before

transformation into electrocompetent *E.coli* for plasmid amplification. pLS-SceI was a gift from Nadav Ahituv (Addgene no. 137725). Sanger sequencing was used to confirm successful construction. The purified plasmid was sequenced (Mi-seq) with custom primers (pLSmP-ass-seq-R1v2 and PLSmP-ass-seq-R2v2; Supplementary Data 23). The association function in the MPRAflow[100] pipeline was used to map unique barcodes for each CRS. A lentivirus library was generated by transfecting HEK293T cells with the plasmid library. After two days, the supernatant was collected and concentrated, and this lentiviral library was used to transduce KMS11, RPMI-8226, and L363 cells in triplicate. After three days, DNA and RNA were harvested, plasmid RNA reverse transcribed, and plasmid DNA and cDNA amplified by PCR, further adding adapters for final NovaSeq sequencing (Illumina). MPRAflow[100] was used to count barcodes and log2 DNA/RNA ratios for each CRS. Activity of ref *vs* alt allele was calculated using MPRAnalyze[101], with CRS direction, barcode, and replicate as covariates. Primers were used as published except those referenced in Supplementary Data 23, which were used to accommodate novel adaptor sequences.

The pre-existing MPRA data for L363 and MOLP8 cells are described in ref. 11. In short; we screened 1039 variants in high LD ($r^2 > 0.8$) with MM lead variants. For each one, we designed twelve 120 bp oligonucleotide sequences corresponding to reference and alternative alleles in six genomic contexts (both strands × three sliding windows with the variant at −20, 0, and +20 bp from the center). Sequences were coupled to a reporter gene with random 20 bp sequence barcodes 3′ of its open reading frame. Following transfection, the transcriptional activity of each construct was measured by determining the barcode representation in reporter mRNA relative to DNA, calculated using MPRAscore[102]. Plasmid sequencing identified $1.73 \times 10^6$ unique barcodes tagging 12,378 (99.2%) of the 12,468 designed oligonucleotides. F

### Luciferase reporter assays

For loci not evaluated by MPRA, we performed luciferase assays. A region surrounding the variant (120 bp or 250 bp) was cloned into luciferase reporter constructs (pGL4.23[luc2/minP] or pGL3 basic; Promega). Constructs and renilla control vectors were transfected by nucleofection with Amaxa (Lonza), using kit V program X-01 for KMS-11 cells; or the Neon electroporation system (Life Technologies) using 2 pulses at 1250 V, 10 ms for Raji; 1 pulse 20 ms 1550 V for U266B1 and 1400 V, 3 pulses, 10 ms for L363. Cells were harvested after 20–24 h incubation at 37 °C, and 5% $CO_2$ and luciferase activity were quantified (DualGlo, Promega E1960). Two technical replicates of each of the three biological replicates were normalized to the renilla control. Biological replicates were mean-centred, and a change in transcriptional activity was calculated as the difference in normalised reads between the reference and alternate alleles. Significance was calculated with a two-sided, paired *t*-test.

### Additional gene expression data sets

To test for enrichment of target gene expression in hematopoietic cell types, we used bulk RNA-seq data for sorted blood cell populations[103], and pseudo-bulked single-cell mRNA-seq data for 35,882 mononuclear blood and bone marrow cells[104].

### Protein quantitative locus (pQTL) analysis

Plasma samples collected from 46,665 UK Biobank participants of European descent were analyzed using Olink (UK Biobank application no. 65851)[105]. The Olink platform consists of 2941 immunoassays targeting 2925 proteins. The measurements were quantile-normalized and adjusted for age, sex, and sample age. Association testing was performed using a linear mixed model[106]. LD score regression was used to account for inflation in test statistics due to cryptic relatedness and stratification[107]. *P*-values were computed using a likelihood-ratio test, and the significance threshold was set to $1.8 \times 10^{-9}$. 24,824 sentinel

trans-pQTLs were discovered after recursive conditional analysis to dissect secondary pQTLs and LD-based clumping[105]. We used SomaScan v4 data for 35,892 Icelanders47 for replication, representing 4907 aptamer-based assays targeting 4719 proteins. The same pipeline was used to derive the lead trans-pQTLs as described for the Olink data. To assess whether the MM variants affect the levels of plasma proteins measured using the Olink platform, we searched for pQTL lead variants that co-localize ($r^2 > 0.8$) with MM risk variants and found pQTLs for 21 proteins (Supplementary Data 17). Six of the proteins associated with more than one MM variant, suggesting a potential causal relationship with MM. To test this, we performed Mendelian Randomization analysis between each of these proteins and MM. Variant effects on proteins coded by IL5RA and BCMA showed significant association with MM risk ($P_{IVW} = 5.6 \times 10^{-5}$ and $P_{IVW} = 9.0 \times 10^{-13}$, respectively) and were therefore investigated further.

### Mendelian randomisation analysis

Two-sample Mendelian Randomisation (2S-MR) was used to examine the causal relationship between leukocyte telomere length (LTL) and pQTLs (exposures) with MM risk (outcome) using the TwoSampleMR package[108,109]. Association data for LTL were obtained from ref. 46. For each variant, effect estimates, and standard errors were retrieved. Variants were considered potential instruments if they were associated at $P < 5 \times 10^{-8}$, minor allele frequency $> 0.01$. To avoid co-linearity, correlated variants were excluded ($r^2 \geq 0.01$). For each variant, causal effect estimates were generated as odds ratios per one standard deviation unit increase in LTL ($OR_{SD}$), with 95% confidence intervals (CIs), using the Wald ratio (Supplementary Data 20). Causal effects were also estimated using a random-effects inverse weighted variance (IVW-RE) model, which assumes each variant identifies a different causal effect. To assess robustness, we compared causal estimates and associated P-values using weighted median (WME) and weighted mode-based (WMBE) methods (Supplementary Data 15). Directional pleiotropy was assessed using MR-Egger regression, and the Steiger test was used to infer the direction of causal effect for exposures (Supplementary Data 16). For this, we estimated the PVE using Cancer Research UK lifetime risk estimates for MM. A leave-one-out strategy under the IVW-RE model was employed to assess the potential impact of outlying and pleiotropic variants (Supplementary Data 21).

### Bayesian test for colocalisation

To test if pleiotropic associations reflect shared variants, we performed colocalization using analysis using Coloc[110] across 1 Mb genomic regions of either side of lead variants of interest. Coloc enumerates four possible configurations of causal variants for two traits, calculating support for each model based on a Bayes factor. Adopting default prior probabilities, a posterior probability $\geq 0.80$ was considered as supporting a specific model.

### CRISPR/Cas9 deletion of variant-harboring regions

To delete the rs4273077 and 4792800-harboring regions in *TNFRSF13B*, we used dual-sgRNA CRISPR/Cas9 genome editing. sgRNA pairs were selected using CRISPOR (crispor.org; Supplementary Data 24) and cloned into the pSpCas9(BB)-2A-GFP PX458 vector (Addgene no. 48138). Cloned sgRNA pairs were co-transfected (ThermoFisher Neon) into Raji cells. After 24 hours, GFP-positive cells were isolated by fluorescence-activated cell sorting. RNA was extracted (RNeasy plus micro kit; Qiagen) and reverse-transcribed. Using TaqMan™ Fast Advanced Master Mix (Applied BioSystems) and PrimeTime qPCR assays (IDT), we quantified the mRNA levels of the two main *TNFRSF13B* transcript isoforms with *ATCB* and *GAPDH* as controls (Supplementary Data 25). To verify deletion efficiency, the targeted regions were PCR-amplified from genomic DNA and analyzed on 2% agarose gels (Supplementary Data 24 and Supplementary Fig. 8).

### Reporting summary

Further information on research design is available in the Nature Portfolio Reporting Summary linked to this article.

## Data availability

Genotyping data have been deposited in Gene Expression Omnibus (GEO) with accession codes GSE21349, GSE19784, GSE24080, GSE2658, and GSE15695[https://www.ncbi.nlm.nih.gov/geo/]; in the European Genome-phenome Archive (EGA) with accession code EGAD50000000422 [https://ega-archive.org/studies/EGAS50000000 292] in the European Bioinformatics Institute (EMBL-EBI) ArrayExpress repository with accession code E-MTAB-362 and E-TABM-1138[https://www.ebi.ac.uk/biostudies/arrayexpress/]; and the database of Genotypes and Phenotypes (dbGaP) with accession code phs000 207.v1.p1[https://www.ncbi.nlm.nih.gov/gap/]. Summary-level GWAS data are available through EGA under accession numbers EGA50000000280, EGAS50000000292, EGAZ50000000827, and EGAZ50000000828[https://ega-archive.org/]. Expression data have been deposited in GEO with accession codes GSE21349, GSE2658, and GSE31161 [https://www.ncbi.nlm.nih.gov/geo/] and in EMBL-EBI ArrayExpress with accession code E-MTAB-2299[https://www.ebi.ac.uk/biostudies/arrayexpress/]. The accession number for the KMS11 ChIP-seq data is EGA: S00001002414[https://ega-archive.org/]. The GM12878 chromatin data is publicly available from UCSC. The sequencing data for the MPRA experiment have been deposited in the Sequence Read Archive, accession no. PRJNA679966. The ATAC-seq data for CD138⁺ MM plasma cells have been deposited in the EGA, accession no. EGAS00001005394 and EGAD00001007814[https://ega-archive.org/]. Publicly available eQTL data from the eQTLGen Consortium[http://www.eqtlgen.org] and gene expression data from the NCBI Gene Expression Omnibus (GEO) repository, accession numbers GSE111199, GSE24759, GSE15695, GSE4581, GSE19784, GSE26760, and GSE5900[https://www.ncbi.nlm.nih.gov/geo/]. Genotype data for the UK Biobank data and the proteomics data can be accessed at https://ukbiobank.dnanexus.com/landing. The UK Biobank Resource was used under application number 65851. The Icelandic genomic data and proteomics data have been described previously[52]. While these individual-level data cannot be shared by Icelandic law, we are open to collaborations, as we have been in the past. The remaining data are contained within the paper and Supplementary Files.

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

## Acknowledgements

This work was supported by grants from the Swedish Research Council (2017-02023 and 2018-00424 to B.N.), the Swedish Cancer Society (200696 to B.N.), the Nordic Cancer Union (R217-A13329-18-S65 to B.N.), European Research Council (EU-MSCA-COFUND 754299 and 847583 CanFaster to B.N., L.D.L., N.U.D., Z.A.), Stiftelsen Borås Forsknings- och Utvecklingsfond mot Cancer (to B.N., U-H.M.), Myeloma UK (to R.S.), Cancer Research UK (C1298/A8362 to R.S.), European Commission Horizon 2020 (856620 to K.H.), the Dietmar Hopp Foundation (to H.G.), National Institute of Health Research and Institute of Cancer Research (to A.S.), Academy of Medical Sciences (SGL024/1013), and Mr. Ralph Stockwell. We are indebted to the patients who participated in the study.

## Author contributions

B.N. and R.H. designed the study. M.W., B.N., and L.D.L. performed bioinformatic analyses. G.H., G.T., L.S., A.S., P.L., M.T., M.P., L.E., N.W., T.O., A.N., A.L.A., and A.L.d.L.P. performed additional bioinformatic analyses. A.G., L.D.L., M.W., and N.U.D. performed the main laboratory experiments. C.C., R.A., M.P., and Z.A. contributed to the experiments. M.H., T.O., T.R., U.T., K.S., A.J.V., A.W., H.G., G.M., I.J., K.H., S.N.S., L.S., S.R., S.Y.K., M.K., U-H.M., T.J.L., P.Su., P.So., E.F., R.H., P.V., U.P-K., F.S., H.T., A.F., C.S., F.v.R., and G.H.E. contributed samples or association data. R.H., B.N., M.W., L.D.L., T.R., G.T., U.T., and G.H. drafted the manuscript. All authors contributed to the final manuscript.

## Funding

## Competing interests

T.O., G.H.H., G.T., L.S., P.Su., E.F., G.H.E., I.J., T.R., U.T., S.N.S., and K.S. are employed by deCODE Genetics/Amgen Inc. The remaining authors declare no competing interests.

## Additional information

[1]Division of Genetics and Epidemiology, The Institute of Cancer Research, London SW7 3RP, UK. [2]Department of Laboratory Medicine, Lund University, SE-221 84 Lund, Sweden. [3]Lund Stem Cell Center, Lund University, SE-221 84 Lund, Sweden. [4]deCODE Genetics/Amgen, Sturlugata 8, IS-101 Reykjavik, Iceland. [5]Department of Internal Medicine V, University of Heidelberg, 69120 Heidelberg, Germany. [6]Landspitali, National University Hospital of Iceland, IS-101 Reykjavik, Iceland. [7]Faculty of Medicine, University of Iceland, IS-101 Reykjavik, Iceland. [8]University Hospital Ostrava and University of Ostrava, Ostrava, Czech Republic. [9]Institute of Experimental Medicine, Academy of Sciences of the Czech Republic, Prague, Czech Republic. [10]Department of Integrative Medical Biology, Umeå University, SE-901 87 Umeå, Sweden. [11]Department of Radiation Sciences, Umeå University, SE-901 87 Umeå, Sweden. [12]Myeloma Center, University of Arkansas for Medical Sciences, Little Rock, AR, USA. [13]Southern Älvsborg Hospital, SE-501 82 Borås, Sweden. [14]Perlmutter Cancer Center, Langone Health, New York University, New York, NY, USA. [15]Department of Hematology, Erasmus MC Cancer Institute, 3075 EA Rotterdam, The Netherlands.

[16]Department of Cancer Research and Molecular Medicine, Norwegian University of Science and Technology, Box 8905, N-7491 Trondheim, Norway. [17]German Cancer Research Center (DKFZ), D-69120 Heidelberg, Germany. [18]MSB Medical School Berlin, Berlin, Germany. [19]Hopp Children's Cancer Center, Heidelberg, Germany. [20]Section of Hematology, Sahlgrenska University Hospital, Gothenburg SE-413 45, Sweden. [21]Skåne University Hospital, SE-221 85 Lund, Sweden. [22]Department of Haematology, University Hospital of Copenhagen at Rigshospitalet, Blegdamsvej 9, DK-2100 Copenhagen, Denmark. [23]Faculty of Medicine in Pilsen, Charles University, 30605 Pilsen, Czech Republic. [24]Broad Institute, 415 Main Street, Cambridge, MA 02142, USA. [25]These authors contributed equally: Molly Went, Laura Duran-Lozano. [26]These authors jointly supervised this work: Richard Houlston, Björn Nilsson.
✉e-mail: richard.houlston@icr.ac.uk; bjorn.nilsson@med.lu.se

