## [Peer Review File · Nature Communications]

Deciphering the genetics and mechanisms of predisposition to multiple myelomaREVIEWER COMMENTS

Reviewer #1 (Remarks to the Author):

This is the largest study to date identifying inherited variants for risk of multiple myeloma (MM). The authors included GWAS data from 9 previously published GWAS (Nat comm 2018). Herein, the authors added UK Biobank study for a total of 10 studies. Across these studies an additional 932 MM cases and 118K controls were added over the prior GWAS for a total of 10,906 cases and 366,221 controls. All individuals were of European Ancestry. The authors perform standard but rigorous analyses of the GWAS data evaluating variants with $MAF > 0.005$ that had high imputation score > 0.8 . The authors also included additional analyses for functional assessment of the MM loci not previously done in their prior 2018 Nat Comm.

The significance of the study is the comprehensive assessment across all MM loci, not just the novel. Evaluations include TWAS, eQTL, epigenomics (e.g., histone marks, methylation), pQTL, precursor conditions, pleiotropy using both publicly generated and study derived data. Extended Figure 1 highlights how the plausible target genes within each locus were identified. I think this figure should be part of Figure 1.

Below are comments to help clarity and interpretation.

Please add to extended figure 1 the cytogenetic bands to represent each loci.

TWAS analysis: please clarify that the TWAS identified 11 loci associated with MM risk but that these loci were the same as that identified in the GWAS.

Table 2 and page 4: It looks like two additional loci (JARID2 and POT1) were not replicated at GWAS threshold. Please clarify definition of MM locus. Also add locus 22q13.1/TOM1 to the table or report the results in text.

Supplementary Tables: please ensure cytoband information (e.g., 2p23.3, 2q21.1, etc) is included in all of the tables to help identify findings across all the reported results.

Figure 4 please add the locus position in the boxes in the figure. Also indicate in legend that these boxes are the 35 MM loci.

Supplementary information on page 2 does not list BRCA2. Please add the MM variant to all the loci reported.

Reviewer #4 (Remarks to the Author):

This study represents a significant contribution to understanding the genetic predisposition of multiple myeloma (MM). By identifying 12 new germline SNPs associated with MM and confirming their driver role through Olink technology, the authors have made a noteworthy advancement in the field.

Overall the paper is well written and has robust in its methods. I have few suggestion and recommendation:

To enhance the impact and citability of this work, I suggest the authors investigate how the identified genetic loci correlate with various serological, genomic subgroups and known drivers of multiple myeloma (MM). For instance, TNFRSF13B and IRF4 have been reported as recurrent structural variant hotspots (Rustad et al., BCD 2020), while POT1 and IRF4 are recurrently mutated (Walker et al., Blood 2018; Maura et al., JCO 2024). Integrating genetic and genomic data would significantly strengthen the paper, providing a more comprehensive understanding of MM genetics and potentially offering insights into personalized treatment strategies.

Reporting the distribution of these loci and variants across different racial groups would be

valuable, especially considering the racial disparities observed in multiple myeloma (MM). Such an analysis could provide insights into the genetic basis of these disparities and contribute to our understanding of MM pathogenesis.

The telomere length is usually shortened in cancer. Why the Authors think that increase LTL increase the risk of MM?

RESPONSE TO REVIEWERS' COMMENTS

We thank the referees for their constructive comments. All points have now been addressed. The changes are listed below and indicated in blue in the revised manuscript.

Response to comments from Reviewer #1

“This is the largest study to date identifying inherited variants for risk of multiple myeloma”, “The significance of the study is the comprehensive assessment across all MM loci, not just the novel. Evaluations include TWAS, eQTL, epigenomics (e.g., histone marks, methylation), pQTL, precursor conditions, pleiotropy using both publicly generated and study derived data.”

Response: We thank the reviewers for their encouragement. We are delighted that they find our article interesting.

“Extended Figure 1 highlights how the plausible target genes. I think this figure should be part of Figure 1”, “Please add to Extended Figure 1 the cytogenetic bands to represent each loci.”

Response: We have upgraded **Extended Figure 1** to a main figure (now **Figure 2**). Cytogenetic bands have been added.

“Please clarify that the TWAS identified 11 loci associated with MM risk but that these loci were the same as that identified in the GWAS.”

Response: Clarified (page 4).

“Table 2 and page 4: It looks like two additional loci (JARID2 and POT1) were not replicated at GWAS threshold. Please clarify definition of MM locus.”

Response: Clarified (page 4).

“Add locus 22q13.1/TOM1 to the table or report the results in text.”

Response: Association testing results for the 22q13.1/TOM1 locus added in the text (page 4).

“Supplementary Tables: please ensure cytoband information (e.g., 2p23.3, 2q21.1, etc.) is included in all tables to help identify findings across all the reported results.”

Response: Cytogenetic bands added.

“Please add the locus position in the boxes in Figure 4. Also indicate in legend that these boxes are the 35 MM loci.”

Response: Locus positions added. Legend revised.

“Supplementary Information on page 2 does not list BRCA2.”

Response: Table of contents corrected (**Supplementary Information**, page 2).

Response to comments from Reviewer #4

“This study represents a significant contribution to understanding the genetic predisposition of multiple myeloma (MM). By identifying 12 new germline SNPs associated with MM and confirming their driver role through Olink technology, the authors have made a noteworthy advancement in the field. Overall the paper is well-written and robust in its methods.”

Response: We again thank the referees for their encouragement. We are pleased that they find our article of interest.

“I suggest the authors investigate how the identified genetic loci correlate with various serological, genomic subgroups and known drivers of multiple myeloma (MM). For instance, TNFRSF13B and IRF4 have been reported as recurrent structural variant hotspots (Rustad et al., BCD 2020), while POT1 and IRF4 are recurrently mutated (Walker et al., Blood 2018; Maura et al., JCO 2024). Integrating genetic and genomic data would significantly strengthen the paper, providing a more comprehensive understanding of MM genetics and potentially offering insights into personalized treatment strategies.”

Response: We have added the suggested references (page 7). Additionally, we now provide data on the relationship between risk alleles and MM subtypes defined by common somatic genetic lesions (new **Supplementary Table 5**). Previous work has uncovered relationships between the risk loci at 11q13.3 and 5q15 with t(11;14) MM and hyperdiploid MM, respectively. For newly discovered loci, we found no evidence for subtype-specific associations, suggesting that these regions may have generic effects on MM risk.

“Reporting the distribution of these loci and variants across different racial groups would be valuable, especially considering the racial disparities observed in multiple myeloma (MM). Such an analysis could provide insights into the genetic basis of these disparities and contribute to our understanding of MM pathogenesis.”

Response: We now provide a table of risk allele frequencies across different populations (new **Supplementary Table 8**), including our study population as well as five super-populations from the 1,000 Genomes compendium. We also provide polygenic risk score distributions for each population (new **Supplementary Fig. 4**). Consistent with the higher incidence of MM in individuals of African or African-American ancestry, we observe higher polygenic risk scores in the African super-population, due to a higher prevalence of several MM risk alleles (page 5).

“The telomere length is usually shortened in cancer. Why the authors think that increased LTL increases the risk of MM?”

Response: No changes requested. As discussed on page 7 and in the Discussion section, we find a positive genetic correlation between MM and LTL. In addition, Mendelian Randomization analysis supports a causal association between increased LTL and an increased MM risk.

REVIEWERS' COMMENTS

Reviewer #1 (Remarks to the Author):

This is well done paper, solid work. The authors addressed my prior concerns.

Reviewer #4 (Remarks to the Author):

The Authors replied to all my comments. I confirm that this paper represents an important contribution to the myeloma community.

Francesco Maura, MD